Diversity improves performance in excitable networks

Gollo Leonardo L. 1 2 leonardo.gollo@qimr.edu.au
Copelli Mauro 3
Roberts James A. 1 2
1 Systems Neuroscience Group, QIMR Berghofer Medical Research Institute , Brisbane, Queensland , Australia
2 Centre for Integrative Brain Function, QIMR Berghofer Medical Research Institute , Brisbane, Queensland , Australia
3 Departamento de Física, Universidade Federal de Pernambuco , Recife PE , Brazil
Perez-Acle Tomas
Electronic publication date: 2016 Apr 25
Publication date: 2016
Volume: 4
Electronic Location ID: e1912
Received 2015 Dec 7; Accepted 2016 Mar 17
Copyright: ©2016 Gollo et al.
Copyright year: 2016
Copyright holder: Gollo et al.
License: This is an open access article distributed under the terms of the Creative Commons Attribution License, which permits unrestricted use, distribution, reproduction and adaptation in any medium and for any purpose provided that it is properly attributed. For attribution, the original author(s), title, publication source (PeerJ) and either DOI or URL of the article must be cited.
License URL: https://creativecommons.org/licenses/by/4.0/

Keywords: Diversity, Criticality, Intensity coding, Nonlinear computation, Sensory systems

Funding: ARC CE140100007 Australian Research Council Dementia Research Development Fellowship APP1110975 CNPq 480053/2013-8 310712/2014-9 Center for Neuromathematics #2013/07699-0 CAPES PVE 88881.068077/2014-01 This work was supported by the Australian Research Council Centre of Excellence for Integrative Brain Function (ARC Centre Grant CE140100007). LLG acknowledges support provided by the Australian Research Council and the Australian National Health and Medical Research Council (Dementia Research Development Fellowship APP1110975). MC acknowledges support of Brazilian agencies CNPq (grants 480053/2013-8 and 310712/2014-9), Center for Neuromathematics (grant #2013/07699-0, S. Paulo Research Foundation FAPESP) and CAPES (PVE 88881.068077/2014-01). The funders had no role in the study design, data collection and analysis, decision to publish, or preparation of the manuscript.

==============================
As few real systems comprise indistinguishable units, diversity is a hallmark of nature. Diversity among interacting units shapes properties of collective behavior such as synchronization and information transmission. However, the benefits of diversity on information processing at the edge of a phase transition, ordinarily assumed to emerge from identical elements, remain largely unexplored. Analyzing a general model of excitable systems with heterogeneous excitability, we find that diversity can greatly enhance optimal performance (by two orders of magnitude) when distinguishing incoming inputs. Heterogeneous systems possess a subset of specialized elements whose capability greatly exceeds that of the nonspecialized elements. We also find that diversity can yield multiple percolation, with performance optimized at tricriticality. Our results are robust in specific and more realistic neuronal systems comprising a combination of excitatory and inhibitory units, and indicate that diversity-induced amplification can be harnessed by neuronal systems for evaluating stimulus intensities.

Introduction

In numerous physical (Dagotto, 2005), biological (Weng, Bhalla & Iyengar, 1999) and social (Silverberg et al., 2014) systems, complex phenomena (including nonlinear computations Gollo et al., 2009) emerge from the interactions of many simple units. Such interactions in a network of simple (linear-saturating-response) units generate nonlinear transformations that give rise to optimal intensity coding at criticality—the edge of a phase transition (Kinouchi & Copelli, 2006; Shew et al., 2009; Chialvo, 2010). However, optimal collective responses often require diversity (Tessone et al., 2006). Clear examples of such optimization can be found in collective sports, business, and co-authorship in which different positions or roles require specific sets of skills contributing to the overall performance in their own way.

Diversity in the nervous system, for example, appears in morphological, electrophysiological, and molecular properties across neuron types and among neurons within a single type (Sharpee, 2014), and also in the connectome (Sporns, 2011), i.e., in how neurons and brain regions are connected. A large body of work has been devoted to show the role of heterogeneous connectivity and network topology in shaping the network dynamics (Fornito, Zalesky & Breakspear, 2015; Misic et al., 2015; Gollo et al., 2015; Gollo et al., 2014; Restrepo & Ott, 2014; Matias et al., 2014; Gollo & Breakspear, 2014; Larremore, Shew & Restrepo, 2011; Rubinov, Sporns & Thivierge, 2011; Honey, Thivierge & Sporns, 2010; Rubinov et al., 2009; Honey et al., 2009; Honey et al., 2007). In particular, for example, in the case of resonance-induced synchronization (Gollo et al., 2014), the presence or not of a single backward connection may define whether synchronization or incoherent neural activity is expected in cortical motifs and networks, which has also been confirmed in a synfire chain configuration (Moldakarimov, Bazhenov & Sejnowski , 2015; Claverol-Tinturé & Gross, 2015).

Crucially, diversity in the intrinsic dynamic behavior of neurons is also fundamental and can shape general aspects of the network dynamics (Vladimirski et al., 2008; Mejias & Longtin, 2012). Such intrinsic diversity reduces the correlation between neurons (Savard, Krahe & Chacron, 2011; Burton, Ermentrout & Urban, 2012; Hunsberger, Scott & Eliasmith, 2014; Metzen & Chacron, 2015) and hence populations, enhancing the information content (Padmanabhan & Urban, 2010) and the representation of spectral properties of the stimuli (Tripathy, Gerkin & Urban, 2013). It also affects the reliability of the network response (Mejias & Longtin, 2012) and its firing rate (Mejias & Longtin, 2012; Mejias & Longtin, 2014). However, the role of the inherent diversity among nodes, which in many systems is at least as notable as the connectivity and network topology themselves, has comparatively remained largely unexplored. In particular, although numerous recent works have focused on optimizing features of criticality for the different network topologies (Haldeman & Beggs, 2005; Kinouchi & Copelli, 2006; Copelli & Campos, 2007; Assis & Copelli, 2008; Shew et al., 2009; Chialvo, 2010; Larremore, Shew & Restrepo, 2011; Shew et al., 2011; Yang et al., 2012; Mosqueiro & Maia, 2013; Gollo, Kinouchi & Copelli, 2013; Haimoviciet al., 2013; Plenz & Niebur, 2014), for convenience identical units are ordinarily assumed and the role of nodal intrinsic diversity on the collective behavior thus remains unexplored.

Here for the first time we analyze the collective behavior at criticality (transition point between active/inactive states) in the presence of diversity in the excitability, which proves to be a crucial factor for the network performance: we show that the task of distinguishing the amount of external input, quantified by the dynamic range, can be substantially improved in the presence of heterogeneity. The influence of non-specialized units improves performance by enhancing the capabilities of both the whole network and of specialized subpopulations. We find that enhanced network response is associated with the proximity to a tricritical regime (critical coupling strength and critical density of integrators—the control parameter for diversity). Away from this tricritcal regime, double and multiple percolation may exist in which the dynamics of the subpopulations can be divided based on the nodal excitability. We show the constructive effects of diversity in excitability given by simple bimodal and uniform distributions, more realistic gamma distributions (see Fig. 1), and the robustness in networks combining excitatory and inhibitory units.

Figure 1 Threshold distributions in random networks.

Threshold θ indicates the minimal number of coincident excitatory contributions required to excite a quiescent unit. (A) Bimodal distribution with 80% integrators (θ = 2). (B) Uniform distribution with θmax=5. (C) Gamma distribution with shape parameter a = 2, and scale parameter b = 1.

Methods

Excitable networks with heterogeneous excitability

Employing a general excitable model [susceptible-infected-refractory-susceptible (SIRS)], we characterize the dynamics and identify the constructive role of diversity in excitable networks and neuronal systems. Node dynamics are given by cellular automata with discrete time and states [0 (quiescent or susceptible), 1 (active or infected), 2 (refractory or recovered)]. Synchronous update occurs at each time step (of 1 ms) obeying the rules: an active node j becomes refractory with probability 1, a refractory node becomes quiescent with probability γ = 0.5, and a quiescent node becomes active either by receiving external input (modeled by a Poisson process with rate h), or by receiving at least θj contributions from active neighbors each transmitted with a probability λ. We considered the stochastic refractory period because it accounts for variations and fluctuations in the recovery dynamics; however, similar results are obtained with a deterministic refractory period with a duration of 2 ms. Diversity is introduced in the threshold variable θj of each node j such that nodes with low threshold require fewer coincidental stimuli, being thus easily and more often excited by active neighbors than nodes with higher thresholds. For concreteness, we used Erdős-Rényi random networks with size N = 5000 and mean degree K = 50 independently generated at each trial. Although each network exhibits its own distinct dynamics, the ensemble average responses are very similar across trials.

Network response

The initial condition for computing the firing rate corresponds to the active state. Nodes receive a strong input (h = 200 Hz) for 0.5 s, followed by a transient period of 0.5 s with the chosen input level (h) before computing the average firing rate of each subpopulation over a period of 5 s. The reported firing rate corresponds to the average over five trials each one utilizing an independent random network.

Mean-field approximation

In the presence of diversity the mean-field map is given by a set of equations for each subpopulation, exhibiting a particular sensitivity to inputs from neighbors (Gollo, Mirasso & Eguíluz, 2012). The dynamics of subpopulations are characterized at the ensemble average level. This mean-field approach represents a substantial reduction in the dimensionality of the system whose dynamics is estimated at the subpopulation level. For each subpopulation with threshold θ, the density of refractory units Rθ at time t + 1 is given by Rt+1θ=Ftθ+1−γRtθ, where Ftθ denotes the density of active units, and γ the recovery dynamics from the refractory state. The evolution of the density of active units follows Ft+1θ=Qtθ1−1−h1−Λtθ, where Qtθ is the density of quiescent units, h is the rate of the Poissonian external driving; Λtθ= ∑i=0θ−1KiλFti1−λFtK−i represents the probability of not receiving at least θ neighboring contributions at time t, where Ft is the weighted average of the density dθ of active units in each subpopulation Ft= ∑θdθFtθ, K is the network average degree, and λ is the synaptic efficacy. Adding to the previous equations the normalizing condition that nodes must be one of the three states at all times, Ftθ+Qtθ+Rtθ=1, we obtain the complete mean-field map: (1) Rt+1θ=Ftθ+1−γRtθ,

(2) Ft+1θ=Qtθ1−1−h1−Λtθ,

(3) Qt+1θ=1−Rt+1θ−Ft+1θ.

Integrating this map (Gollo, Kinouchi & Copelli, 2012), we find the stationary distributions (Fθ) for each subpopulation, which are compared with the simulation results.

Gamma distribution

The discrete gamma distribution of thresholds is given by the smallest following integers drawn from the probability density function f(θ) = θa−1e−θ∕b(baΓ(a))−1, where a and b are shape and scale parameters, respectively.

Results

Mix of specialized and nonspecialized nodes outperforms either alone

To understand the role of diversity in the excitability of nodes we start with the simplest case, a discrete bimodal distribution, in which half the units are so-called integrators with θ = 2, and the other half are nonintegrators with θ = 1. In the presence of weak external driving (h = 10−2 Hz), the most excitable units (in red with θ = 1) fire more often than the integrators (in blue), as depicted in Fig. 2, and the dynamics of such networks depends on the coupling strength λ. For weak coupling (Fig. 2B), the most excitable units fire at a relatively low rate while the integrators are nearly silent, firing only sparsely. However, we shall see that their small contribution can play a major role in the network response to varying external stimuli. Increasing the coupling (Figs. 2C and 2D), both subgroups fire more often but the firing rate of the integrators remains much lower than the nonintegrators (Fig. 2A) and the network dynamics can be essentially split in two clusters that interact, albeit weakly.

Figure 2 Illustrative dynamics in the simplest case of diversity in the excitability threshold θ.

Bimodal distribution with equal numbers of integrators (θ = 2) and non-integrators (θ = 1). (A) Spontaneous activity F(h = 0) ≡ F0 versus coupling strength λ. The critical coupling for the subpopulation θ = 1 and θ = 2 is respectively λc1=0.0425 and λc2=0.0675. (B–D) Time traces and raster plots for different coupling strength λ. Top panels: instantaneous firing rate ρ averaged over nodes from each subpopulation (θ = 1 is in red, θ = 2 is in blue). Bottom panels: raster plot of 500 randomly chosen units from the integrator (blue) and nonintegrator (red) subpopulations. The external driving is h = 10−2 Hz.

Figure 3 A specialized subpopulation with increased coding performance emerges with diversity.

(A) Response curves (mean firing rate F versus stimulus rate h) for the subpopulations of θ = 2 (blue), θ = 1 (red), and the whole network (gray). Variables F0.1 and F0.9 (red dashed lines), and h0.1 and h0.9 (black arrows) are used to calculate the dynamic range Δ1 (red arrow) for the subpopulation with θ = 1, where Fx=F0+xFmax, hx is the corresponding input rate to the system, and F0 is the firing rate in the absence of input. Solid black lines correspond to the mean-field approximation (see ‘Methods’). (B) Dynamic range Δ is optimized for different coupling strengths λ for the two subpopulations. Dotted lines connect the numerical data points. Inset: susceptibility χθ for the two corresponding subpopulations; susceptibility maxima coincide with the peaks of the dynamic range. Susceptibility was calculated over 500 trials of 100 ms after transients of 0.5 s.

Our main analysis focuses on the input–output response function of networks subjected to external driving h, whose intensity varies over several orders of magnitude, as is commonly observed in sensory systems, for example. Response functions F are defined as the mean activity over 5 s of the whole network or a subset thereof with the same threshold θ (Fig. 3A). F curves exhibit a sigmoidal shape with low output rates for weak stimuli and high rates for strong stimuli. Aside from the saturated region, the subpopulation of integrators (blue) fires less than the subpopulation of nonintegrators (red), and (for this particular distribution) the activity of the whole network (gray) corresponds to the average between the two subpopulations. For the two subpopulations as well as for the whole network our mean-field approximation is capable of reproducing the response functions remarkably well. From the shape of the response functions we quantify the range in which the amount of input can be coded by the output rate (Fig. 3A). This dynamic range Δ = 10log10(h0.9∕h0.1) is a standard measure (Kinouchi & Copelli, 2006) that neglects the confounding ranges of too small sensitivity [top 10% (F > F0.9) and bottom 10% (F < F0.1)], and quantifies how many decades of input h can be reliably coded by the output activation rate F (see caption of Fig. 3A for further details).

Figure 4 Threshold diversity improves performance.

Comparison of dynamic ranges for homogeneous networks where all units have threshold θ = 1 (green, Δ1homo) or θ = 2 (purple, Δ2homo) with the θ = 1 subpopulation of the bimodal distribution (red, Δ1). Solid black lines correspond to the mean-field approximation (see ‘Methods’), dotted lines join the numerical data points.

Although isolated units (λ = 0) code input intensity very poorly (small Δ), increasing the contribution from neighbors (by increasing the transmission probability λ) substantially enhances the dynamic range (Figs. 3B and 4). However, this occurs only for coupling smaller than a critical value λc, at which a phase transition to self-sustained activity occurs (e.g., Fig. 2A). As the coupling strength increases beyond the critical value, the dynamic range decays because the effective output range is reduced by increasing levels of self-sustained activity (Kinouchi & Copelli, 2006). Since our mean-field approximation exhibits good agreement with the numerical results for the response functions, the dynamic range is also precisely captured. There is only one outlier point that corresponds to the critical point. At criticality the growth of the response functions is abnormally slow (anomalous exponent) causing a substantial enhancement of the dynamic range that cannot be matched by mean-field approximations (Gollo, Kinouchi & Copelli, 2012). In this simple bimodal case the phase transition occurs at different λ values for the two subpopulations, evidenced by peaks of the dynamic range Δθ as well as the susceptibility (Fig. 3B and its inset). The susceptibility captures the variability of the instantaneous ensemble firing rate around its mean value (over time) for each subpopulation, and it is formally defined as χθ≡ρθ2∕〈ρθ〉−〈ρθ〉, where ρθ = Fθ(h = 0). The critical value of the coupling (curve’s peak) for Δθ is larger for integrators than for nonintegrators. Moreover, as evidenced by the difference between the maximum dynamic range of each subpopulation (Δmax1−Δmax2≃15 dB, Fig. 3B), nonintegrators greatly outperform integrators.

In the presence of diversity the specialized subpopulation of nonintegrators (Δ1) outperforms the two extreme cases with no diversity (homogeneous distribution) in which all units are either integrators Δ2homo or nonintegrators Δ1homo (Fig. 4). This happens because the response of the specialized units improves when they can also take advantage of the contribution of the other subpopulation of integrators, which require simultaneous neighboring stimulation to be effective. In the presence of integrators the network requires stronger coupling to switch to the active state. Therefore, due to a stronger coupling, the amplification of weak stimuli at criticality and thus the dynamic range are greater than in the absence of diversity. Remarkably, however, having all nodes behave like the specialized ones impairs performance.

Tricriticality optimizes coding performance

Henceforth, since criticality optimizes performance, we focus on characterizing the critical behavior for various types of diversity in the excitability. Varying the density of integrator units (with θ = 2) while the rest are nonintegrators, we find a critical point separating two regimes (Fig. 5A): for a low density of integrators (green region) the phase transition to the regime of spontaneous activity is continuous (transcritical bifurcation in the mean-field equations for the model, see Methods); for a high density of integrators (purple region) the phase transition to the regime of spontaneous activity is discontinuous (saddle–node bifurcation in the mean-field equations) (Gollo, Mirasso & Eguíluz, 2012). The presence of two different critical couplings in the region with continuous phase transitions indicates double percolation, where the most excitable units percolate for a weaker coupling than integrators. The critical-coupling curves (λc) grow with the density of integrators for both the subpopulation of integrators (blue) and nonintegrators (red) and these curves collapse at the tricritical point (orange line). This collapse is also captured by the mean-field approximation because the critical regions can be detected with good precision. As represented in the inset of Fig. 5A, the tricritical point corresponds to a critical density of integrators (d = 0.8) separating regions undergoing continuous and discontinuous phase transitions. At this transition, apart from a collapsing of critical-coupling curves, the maximum susceptibility also changes qualitatively (Fig. 5B). The inset of Fig. 5B illustrates the curves of susceptibility for the subpopulation of θ = 1 for different densities of integrators; the susceptibility curve becomes more sharp for discontinuous transitions. Strikingly, as shown in Fig. 5C, the intermediary regime with a density of integrators of 80% (orange line) poised between the regions of continuous and discontinuous phase transitions yields optimal performance. In other words, the maximum dynamic range for generalized bimodal distributions occurs at the tricritical point. The inset of Fig. 5C shows the response functions corresponding to the tricritical regime. In this regime the sensitivity is more than two orders of magnitude larger than in the absence of diversity (Δ1homo in Fig. 4).

Figure 5 Performance is optimized at tricriticality with a critical density of integrators and a critical coupling strength.

General bimodal distribution with varying densities of integrators (θ = 2) and non-integrators (θ = 1). (A) Critical coupling strength (λc) as a function of the density of integrators (d) for the two subpopulations. Curves collide at a tricritical point (orange line), separating regimes with continuous (2nd order, green) and discontinuous (1st order, purple) phase transitions. Solid black lines correspond to the mean-field approximation. Inset: spontaneous activity F0 versus coupling strength λ for the critical density of integrators. (B) Maximum susceptibility χmax versus density of integrators. Inset: susceptibility of subpopulation with θ = 1 versus coupling strength for three integrator densities (0.75, 0.8, 0.85). (C) Maximum dynamic range Δmax versus density of integrators. Inset: response curves at the tricritical point (λ = 0.1075).

Diversity can yield multiple percolation

Large dynamic ranges also occur at criticality in other distributions such as the uniform distribution. In this case, the number of units with threshold θ is evenly distributed between 1 and θmax, as depicted in the top panel of Fig. 6 for an exemplar case with θmax=5. Notably, for the uniform distribution, Δmax1 is much greater than Δmax of the other subpopulations (Fig. 6A) and of the whole network (inset).

Figure 6 Multiple percolation and optimal performance in uniform distributions of thresholds.

(A) Maximum dynamic range Δmax versus the maximum threshold of the uniform distribution θmax for each subpopulation, and the whole network (inset). (B) Critical coupling strength (λc) as a function of θmax for each subpopulation. The whole network (inset) exhibits two peaks for θmax>3. (C) Susceptibility versus coupling strength for each subpopulation, and the whole network (inset). Arrows at the bottom of the panel identify the critical couplings.

In contrast to the bimodal distribution (Fig. 5A), the critical coupling curves of the subpopulations for the uniform distribution grow with θmax without collapsing (Fig. 6B). Hence, the system exhibits multiple critical couplings. However, the network taken as a whole exhibits at most two peaks of susceptibility (insets of Figs. 6B and 6C). As shown in the inset of Fig. 6B, the lowest-threshold critical coupling for the whole network matches the critical value for the subpopulation with θ = 1, and the other reflects the contribution of all subpopulations. Figure 6C displays the susceptibilities for each subpopulation and the whole network (inset). The larger the θ of the subpopulation, the greater the coupling required to optimize the susceptibility, leading to a subpopulation hierarchy.

More details about multiple percolation in the case of θmax=6 are also given in Fig. 7. Figure 7A shows how the self-sustained activity grows in each subpopulation and in the whole network as a function of the coupling strength. Each subpopulation has a different percolation threshold. This is also clear from the peaks of the derivatives of the self-sustained activity with respect to λ (Fig. 7B). Another key feature is that the peak corresponding to the subpopulation of θ = 1 is much higher than the others, which is analogous to the shape of χ shown previously in Fig. 6C. Both the time traces of ρ and the raster plot for a near critical coupling of the most excitable subpopulation (Fig. 7C) show large fluctuations in the activity of this subpopulation but only minor activity in the other subpopulations. The active recruitment of other subpopulations requires stronger coupling, as shown in Figs. 7D–7F.

Figure 7 Multiple percolation for a uniform distribution of thresholds with six subpopulations.

(A) Firing rate for each subpopulation and for the whole network (gray curve) as a function of the coupling strength in the absence of input. (B) Derivatives of curves of A with respect to λ with step size Δλ = 0.005. Labels C, D, E, and F indicate the coupling strengths used in the correspondingly-labeled panels. (C–F) Time traces and raster plots for different coupling strength λ. A, B, C: instantaneous firing ρ averaged over nodes from each subpopulation and the whole network (gray). D, E, F: raster plot of 1,000 randomly chosen units. Nonintegrator units are in red and other colors represent integrator units with different thresholds (see C). The external driving is h = 10−2 Hz.

Extending to a more realistic distribution

The gamma distribution is more general and presumably more realistic than the bimodal and uniform distributions. As presented in the Methods and illustrated in Fig. 1, it is described by two independent parameters, shape a and scale factor b, and generalizes the exponential, chi-squared, and Erlang distributions. Exploring random networks with thresholds given by discrete gamma distributions, we find large dynamic ranges (Fig. 8). The maximum dynamic range for both the subpopulation with θ = 1 and the whole network can reach ∼40 dB (Figs. 8A–8C). For some gamma distributions, the dynamic range of the whole network can be larger than the specialized subnetwork of Δmax1 (dark gray area of Fig. 8C). In these particular cases Δmax1 is very small; the maximum dynamic range of this subpopulation is at its floor value, similar to the uncoupled case with λ = 0 (see e.g., Fig. 4), implying that the network contribution to its dynamic range is negligible. A more detailed picture of the dynamic range of the subpopulations reveals that very different relations among subpopulations may appear for small changes in the parameters controlling the discrete gamma distributions (see inset of Figs. 8D–8F). For a reasonably large density of non-integrators and a distribution spanning from θ = 1 to large thresholds (Fig. 8D), Δmax1 largely exceeds the other subpopulations, and the maximum dynamic range decays with threshold. For a fixed scale parameter b, increasing the shape parameter a, the number of non-integrator units decays and Δmax1 suffers a large reduction (Fig. 8E). Moreover, a further increase in the shape parameter a leads to a regime in which the dynamic range grows with θ until its maximum value and then decays for larger thresholds (Fig. 8F). This regime is significant because the dynamic range of the whole network (horizontal line) outperforms all subpopulations. Although it is often taken as a basic truth that the whole is greater than its parts, we find that this is not a general rule for all complex systems. Among all considered distributions (bimodal, uniform, and gamma) we only find the whole network outperforming all subpopulations in a small region of parameters of the gamma distribution in which the subpopulation with θ = 1 cannot benefit from network interactions (top-right pale region of Fig. 8A).

Figure 8 Optimal performance for gamma-distributed thresholds: the whole can outperform its parts.

(A–C) Maximum dynamic range versus shape parameter a and scale parameter b of the gamma distribution. (A) Specialized (with highest sensitivity) subnetwork; (B) the whole network; (C) difference between the whole network and the specialized subnetwork. (D–F) Maximum dynamic range for various subpopulations and the whole network (horizontal gray line). Inset: gamma distribution of threshold values for the corresponding shape and scale parameters.

Networks with excitatory and inhibitory nodes

Our main result that performance can be substantially enhanced with diversity is also robust with respect to the presence of inhibition. We introduced bimodal diversity in the thresholds as follows. First, we fix the proportion of inhibitory units at 20%. For each total density of integrators, we distribute these according to three simple cases: (i) all inhibitory units are integrators (thus requiring a total integrator density d ≥ 20%, with the excitatory units comprising the d − 0.2 integrators and the remainder nonintegrators); (ii) all inhibitory units are nonintegrators (thus requiring a total integrator density d ≤ 80%); and (iii) diversity in the threshold of the inhibitory units (fixed at 50% integrators and 50% nonintegrators, thus requiring a total integrator density 10% ≥d ≤ 90%). This covers the two extreme cases (i) and (ii) and an intermediate case (iii). After an inhibitory neuron spikes, post-synaptic quiescent neurons receive inhibition with probability λ. Upon arrival, inhibition prevents the unit from spiking within a time-step period irrespective of the number of excitatory active neighbors (i.e., we model shunting inhibition). We find that inhibition has two effects on the response function, influencing the dynamic range in opposite ways. On the one hand, inhibition prevents a rapid increase in the firing rate for small input. On the other hand, it prevents saturation for large input. The first effect tends to reduce the dynamic range whereas the second effect tends to increase it.

In the absence of diversity, the overall effect reported in the literature is a small reduction in the network dynamic range (Pei et al., 2012). In the presence of diversity, however, we find the overall effect counterbalanced and inhibition does not alter the diversity-induced enhancement of Δ. Figure 9 shows the robustness of the maximum dynamic range in the presence of inhibition. Regardless of whether the inhibitory units are integrators (pentagons), nonintegrators (triangles), or a mix of both (square) the dynamic ranges are very similar to the case without inhibition (filled circles). Although inhibition has been shown to crucially shape the network dynamics (Larremore et al., 2014), and diversity in excitatory and inhibitory populations may have different effects (Mejias & Longtin, 2014), we found that in the presence of diversity inhibition produces only minimal quantitative differences in the coding performance of networks.

Figure 9 Robustness of optimization in a network with 20% inhibitory units.

Thresholds are drawn from a bimodal distribution of integrators (θ = 2) and non-integrators (θ = 1). Maximum dynamic range versus coupling strength for the specialized subnetwork (A) and the whole network (B) for different types of inhibitory units: nonintegrating (triangles), integrating (pentagons), half integrating and half nonintegrating (squares), and the case without inhibition (filled circles).

Discussion

Minimal models play a key role in elucidating the mechanisms and dynamics of complex systems. Following this approach and investigating the impact of diversity in the intrinsic excitability, we have shown that: (i) Diversity offers clear-cut advantages in distinguishing input with respect to homogeneous networks; (ii) At the tricritical point the system benefits from multiple critical instabilities, thereby optimizing performance; (iii) Subpopulations percolate in order of decreasing excitability; (iv) The collective response from the entire network can outperform all subpopulations but only when the specialized subpopulation is underrepresented in the distribution of thresholds; (v) The main results are robust to more realistic distributions, and can be applied to cortical systems composed of excitatory and inhibitory neurons.

Diversity has been a keystone of the recruitment theory that proposed the first explanation for how animals can distinguish incoming input spanning many orders of magnitude, even when each individual sensory neuron distinguishes only a narrow dynamic range (Cleland & Linster, 1999). The proposed mechanism there requires many neurons exhibiting responses tuned to specific (short) ranges of input but with the ensemble of specific ranges spanning several orders of magnitude. The limitation of this recruitment mechanism is evident because neurons would need to have receptor densities also varying across orders of magnitude, which is not found experimentally (Chen & Yau, 1994; Cleland & Linster, 1999). A competing hypothesis claims that diversity is not required, but instead nonlinear interactions are sufficient for sensory systems to cope with incoming input varying over many orders of magnitude (Kinouchi & Copelli, 2006; Copelli, 2007). Our revisited version of the recruitment theory reconciles the two proposals by employing the key ingredient of each one: mutual (non-linear) interactions, which amplify the dynamic range of isolated neurons, and intrinsic diversity in the excitability, which requires only small variability in threshold (and not variations of orders of magnitude as in the classic recruitment theory Cleland & Linster, 1999). Therefore, by showing that diversity enhances the dynamic range of response functions, we establish a revisited recruitment theory with firmer biological plausibility.

Although we have focused on a specific task of distinguishing stimuli intensity, sensory systems also need to handle various other features. As a byproduct and another advantage of diversity, nonspecialized units may execute and specialize in other functions. For example, as recently reported in the moth olfactory system (Rospars et al., 2014), a concurrent function of the detection of stimulus intensity is the ability to respond promptly to external stimuli. Under evolutionary pressure, the ability to execute such complementary functions likely takes advantage of diversity to improve performance.

Our work provides predictions that may be used to guide experiments: (i) Manipulating the coupling strength between neurons (such as done by Shew et al., 2009) should change their dynamic range. Weaker coupling favors units with larger sensitivity and stronger coupling favors units with lower sensitivity. Provided that a variation in coupling strength is large enough, it should be possible to change which subpopulation is most sensitive. (ii) The dynamic range can be substantially reduced if diversity is compromised, for example by neurodegeneration or in genetically modified animals. Another possibility for manipulating diversity would be to induce changes in a specific targeted population. Although these predictions may be challenging to test experimentally, the numerical results presented here will aid in narrowing down the proposed questions.

The importance of having diversity in groups is a widely-accepted strategy for improving performance (Jackson & Ruderman, 1995; Van Knippenberg & Schippers, 2007; Joshi & Roh, 2009; Freeman & Huang, 2015). The common examples of businesses, scientific collaborations and sport teams assume that the collective output is enhanced because different elements contribute by providing complementary expertise. Here, however, we focus on a different problem in which all elements are responsible for the same task, but some elements can perform better than others because of their different sensitivity. It is natural to assume that a group of high-sensitivity specialized elements would lead to the optimal outcome. Counter-intuitively, our results show that optimal performance requires a group of diverse elements, including specialized units with high sensitivity and supporting units with low sensitivity. The supporting units do not typically outperform the specialized units but they play a major role in enabling the enhancement of the specialized units’s performance. In the simplest case of diversity (bimodal distribution) the specialized units are the gullible units and their performance is greatly enhanced by the interaction with other prudent units. This combination of units delays the critical transition and provides additional sensitivity in distinguishing stimulus intensity to the specialized units. In fact, although a mixture of these two types of units is beneficial with respect to the unimodal case (with no diversity), there is an optimal recipe for combining them: maximal amplification in coding performance occurs for a critical balance of the two types of units. Such balance corresponds to a critical regime that splits the dynamics in two. Adding extra integrators makes the phase transition discontinuous, and removing them makes the phase transition continuous.

We have demonstrated the benefits of diversity at criticality for different simple distributions of excitability (as requested in the recent literature; Baroni & Mazzoni, 2014). In the context of diversity-induced resonance (Tessone et al., 2006), in which diversity plays a role similar to the noise in stochastic resonance, the firing rate modulation by heterogeneity causes an optimal correlation response of the network to an oscillatory external driving (Mejias & Longtin, 2012; Mejias & Longtin, 2014) but no specific attribute has been previously identified to the network at the optimal level of diversity to justify its optimized response. Addressing this issue, for the first time we provide evidence that the well-known advantages of criticality (Plenz & Niebur, 2014) are magnified at tricriticality. The optimal performance in the simple case of two type of units is found at a tricritical point with a critical coupling separating the active/inactive phases and a critical density of integrators separating the regimes of continuous/discontinuous phase transitions. Even though a continuous phase transition has been proposed for the brain (Chialvo, 2010), hysteresis and multistability observed in models (Gollo, Mirasso & Eguíluz, 2012; Wilson & Cowan, 1972) and experiments (Kastner, Baccus & Sharpee, 2015) suggest that discontinuous phase transitions may also play a functional role.

The dynamics of excitable networks exhibits two regimes: percolating (active phase) and non-percolating (inactive phase) (Saberi, 2015). In the non-percolating regime the coupling is not strong enough to guarantee self-sustained propagation of activity. The network activity eventually dies at the absorbing state and an external stimulation is required to generate a spike. In contrast, the percolating regime is characterized by ceaseless activity. Between these two regimes there is a phase transition. In the presence of diversity (even in the simplest case of bimodal distribution), if the density of integrators is below a critical density, double percolation is observed. The most excitable units undergo a phase transition to the active phase for weaker coupling than integrators. This process is analogous to the recently shown (Colomer-de Simón & Boguñá, 2014) double percolation, which occurs in core–periphery networks with sufficient clustering: core nodes percolate earlier than peripheral nodes as edges are added to the network. Our results also generalize double percolation to arbitrarily high-order multiple percolation, with subpopulation thresholds following a hierarchy of excitability. This analysis shows that the intrinsic properties of the nodes can play a crucial role in disentangling the network activity even in the absence of special topological features such as a core–periphery structure.

Features of the network structure can also play distinctive roles in shaping the dynamic range. Networks in which hubs (nodes with large degree) are mutually connected (i.e., assortative networks) exhibit larger performance than if they were not connected (De Franciscis, Johnson & Torres, 2011; Schmeltzer et al., 2015). In scale-free networks, nodes with higher degree have larger dynamic range (Wu, Xu & Wang, 2007), and the dynamic range of networks grows with the degree exponent, which means that the dynamic range is larger for scale-free networks with more homogeneous degree (Larremore, Shew & Restrepo, 2011). Such distinct roles for diversity in the network connectivity as opposed to diversity in the intrinsic dynamics highlight their difference in nature. Although we have focused on a simple random network topology, the combination of diversity at the unit level with the network level appears to be a rich avenue for future work.

We introduced diversity into our networks by having the node excitabilities follow simple distributions. Our results remained robust in moving from simple distributions to more complex cases, suggesting that the effects of diversity are general and widely-applicable. Another crucial feature of many systems is the presence of excitatory and inhibitory units. Our results are also robust with respect to the presence of inhibition regardless of whether inhibitory units are homogeneous or not. This result indicates that inhibition does not play a large role in the coding performance of the network, which contrasts with its crucial role in other functions such as maintaining the self-sustained activity in the network (Larremore et al., 2014). The reason for such a robustness is that the two effects of inhibition in the response function are opposite and compensate one another: it reduces both the sensitivity to small stimuli and the saturation to large stimuli. This robustness suggests that either diversity in inhibitory neurons (observed in experiments, Whittington & Traub, 2003) should have other functional roles (Mejias & Longtin, 2014) and does not contribute significantly to the network’s coding performance, or that a more complex and detailed modeling approach (Harrison et al., 2015) is needed to address the role of diversity within the subset of inhibitory neurons for coding performance.

Supplemental Information

Supplemental Information 1 High-performance C code

The code to simulate the system dynamics is also freely available at: http://www.sng.org.au/Downloads.

Click here for additional data file.

We would like to thank Jorge F. Mejias for valuable comments and a careful reading of the manuscript.

Additional Information and Declarations

Competing Interests

Author Contributions

Data Availability

The authors declare there are no competing interests.

Leonardo L. Gollo conceived and designed the experiments, performed the experiments, analyzed the data, contributed reagents/materials/analysis tools, wrote the paper, prepared figures and/or tables, reviewed drafts of the paper.

Mauro Copelli and James A. Roberts conceived and designed the experiments, contributed reagents/materials/analysis tools, wrote the paper, reviewed drafts of the paper.

The following information was supplied regarding data availability:

The code to simulate the system dynamics is freely available at:

http://www.sng.org.au/Downloads and a copy of the high-performance C code is provided as Supplemental Information 1.

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
