# Peer review of "Diversity improves performance in excitable networks"

_PeerJ, doi:10.7717/peerj.1912_

## Round 0.1 · original submission · Major Revisions

Thanks a lot for considering PeerJ as the source of publication for your work. As you will see in the following sections, both reviewers agreed that, despite your work coulee be interesting for PeerJ readers, some important issues should be addressed.

·

Basic reporting

Although the findings are interesting and well sustained, my major concern is with their novelty. Diversity in neural networks has indeed not heavily studied, and its role to network function remains to be clarified, but there have been some recent works by other authors, more than what is mentioned in the paper (In other words the issue has been studied more than what the authors want the readers to think). In particular, I find it difficult to see what is novel in the present work compared to that of Mejias & Longtin (2014), already cited in the paper. Although the models are different (mainly continuous versus discrete) the conclusions are at first glimpse very similar: heterogeneity flattens the f-I curve, meaning a larger dynamic range. If there is some novelty in the details of the results or the analyses, then this should clearly stated and thoroughly discussed. Also, it should be clearly stated why it is worth revisiting an apparently settled issue with a discrete model approach.
Some other works to be cited and discussed are: Hunsberger et al., Neural Comput 2014; Savard et al., Neuroscience 2011; Metzen & Chacron, J Neurosci 2015, Mejias & Longtin, Phys Rev Lett 2012.

The Mean-Field results are poorly described in the Results section, and there is no actual conclusion drawn from them. A better description of the mean-field approach as well as a comparison of its results with the simulation results are needed. Otherwise I suggest these results to be removed from the Figures and Methods section.

Raw data has not been made available. For this type of work, I expect model codes to be deposited in an appropriate repository.

Experimental design

No comments

Validity of the findings

* Line 242 says: “(iv) The collective response from the entire network can outperform all subpopulations but only when the specialized subpopulation is far from its optimal tuning”. I don't see this clearly. From most of the results, it appeared to me that the network outperforms non-integrators depending on the distribution of thresholds rather than the unit's tuning. Please clarify.

* Raw data has not been made available. For this type of work, I expect model codes to be deposited in an appropriate repository.

Additional comments

Some minor comments:

* The critical value \lambda_c, mentioned for the first time in line 122, should be marked in Figure 2a with an arrow or similar symbol.
* The susceptibility (line 126) must be explained in plain words, similar to how the dynamic range is explained.
* The concept of percolation should also be explained and its importance discussed.
* In Figure 8a, only the specific case in which the network outperforms the subpopulations is shown. An additional figure, showing the most relevant case of the non-integrators outperforming the network, could also be shown.
* The sub-subsection of Networks with Inhibitory nodes (lines 213) should be a section with a higher hierarchy, i.e., similar to the other sections. In other words, the two sub-subsections under 'More realistic scenarios' are too different to be considered as being part of one subsection.
* In the networks with inhibitory nodes it is said that 20% of units are inhibitory. In the heterogeneous network, how is this 20% distributed among the different threshold populations? 20% of each?
* In the first lines of the section describing networks with inhibition (lines 214-222), it s not completely clear which statements are findings of the present work and which are previous findings. The fact that the inhibitory mechanism of the model is explained later (starts in line 222), made me think that all the previous lines are all previous evidence. Is this correct? Please add some more references or change the writing style to avoid confusion.

Reviewer 2 ·

Basic reporting

The article is very well written, easy to follow and understand, with a thorough literature review. The figures are also very clear, complement the text and are also easy to understand. As it is, the only defect I find in this section is the unavailability of the networks used in the study.

Experimental design

With respect to the experimental design, I found several flaws that require further attention.
1.- The authors do not specify if they used the same network architecture (connectivity between neurons) and only altered the types of neurons or if they generated a random network for each network with different composition.
2.- It is also not specified how many times did the authors generated a random network or if they did it only once. This is an important aspect since different connectivity between nodes can produce different network behavior. In other words, are the results shown in the article the product of averaged performances from several replica of the networks or obtained from the same single random network?
3.- In lines 71-72 in page 3, the authors claim their results generalize to other sizes, connectivities and topologies. This strong affirmation is not proved or demonstrated anywhere, and making it should require prove, at least from my point of view. To prove it, several replica of different random network models generated with different approaches should be tested together with different network sizes and connectivities.

Validity of the findings

Even the results are sound, without addressing the flaws in the experimental design, they lack validity. This is highly relevant specially with respect to the statistical significance of the reported findings, since without several replica of the random network generation it is not possible to calculate.

---

## Round 0.2 · accepted · Accept

Thanks to the authors of this manuscript for your careful addressing of the issues raised by the reviewers. Despite the paper has been dramatically improved since the last version, still some minors issues were raised by reviewer 1. Please read them below and proceed accordingly. Once again thanks a lot for choosing PeerJ as the source of publication of your best work.

·

Basic reporting

No Comments

Experimental design

No Comments

Validity of the findings

No Comments

Additional comments

I think the authors have satisfactorily addressed the concerns raised. There are only minor points that should be corrected or commented:

- It is unclear how the critical coupling for the integrators is defined (Fig 2a), given the smooth (second order) transition to sustained activity. This is very important for Figure 5, where \lambda_c is plotted and several conclusions are drawn.
- The continuous gamma distribution is used to obtain discrete values (threshold). Just to be more statistically rigorous, I suggest to use a discrete equivalent. Although I am no expert on probability distributions, probably the negative binomial will serve the purpose, and there will be parameters equivalent to those that were used with the gamma.

Minor observations:
- The layout of Figure 2 has to be improved, correcting the centering of panels. Also consider that panel 2a is the last one to be mentioned.